# Union-of-Experts: Experts in Mixture-of-Experts are Secretly Routers

## Abstract

Mixture-of-Experts (MoE) is a foundational architecture in modern large language models (LLMs). However, a structural limitation has been overlooked: the router is external to the experts, rendering it unaware of their internal capabilities. This gap between routing decisions and expert capabilities limits model performance. In this paper, we demonstrate that the activations of a small subset of "routing neurons" within each routed expert's own parameters can faithfully capture the match between the expert's capabilities and input tokens. Collectively, these distributed routing neurons within each routed experts compose an implicit, capabilities-aware "router", where the norm of the routing neurons' activations suggests its corresponding expert's weight. A straightforward implementation of this design requires activating all experts to compute these routing signals, where the unselected experts' routing neurons are abandoned. To avoid the computational waste from activating unselected experts, we introduce another novel design: we unify the routing neurons of all routed experts to form a virtual shared expert, replacing the standard shared expert in MoE. In this virtual shared expert, activations are not wasted, as they serve not only for routing but also contribute to the final outputs of both the shared expert and partial of routed experts. We name this new MoE variant Union-of-Experts (UoE), drawing an analogy where the routing neuron acts as each expert's representative, and the virtual shared expert is their union, enabling the experts' autonomous selection and joint statement. We pre-train language models ranging from 1B to 3B parameters, showing that UoE consistently outperforms strong MoE baselines with comparable efficiency.

## 1 Introduction

Mixture-of-Experts (MoE) has garnered increasing research interest. A number of MoE-based LLMs have been proposed in recent works (DeepSeek-AI et al., 2025; Yang et al., 2025; OpenAI, 2025), exhibiting strong performance across a broad spectrum of downstream tasks. In Transformer-based MoEs, the feed-forward network (FFN) is replaced with multiple smaller expert networks, and a router dynamically routes each input token to a subset of experts. This sparse activation mechanism facilitates the training of trillion-parameter models with feasible computational overhead, establishing MoE as a fundamental architecture in modern large language models (LLMs).

However, there is a gap between routing decisions and expert capabilities. Because the router is a standalone module external to the experts, it can only infer their abilities through trial and error. When a token is inappropriately routed, the expert has to adapt to that token, compromising its specialization. To solve this, the "expert autonomy" concept has been proposed in AoE (Lv et al., 2025), wherein all experts process the token and the one with the largest activation norm (indicating the best match) is selected. While this concept improves performance, it incurs a significant computational overhead as the number of experts grows. This inefficiency contradicts the core efficiency goals of MoE models and thus limits the practical deployment of this concept in LLMs, especially under the trend of expanding total expert numbers of industrial MoE models (OpenAI, 2025; Team et al., 2025).

In this paper, we propose **Union-of-Experts (UoE)**, a new MoE architecture that adopts the principle of expert autonomy to achieve satisfactory performance, while maintaining efficiency comparable to traditional MoE models. Figure 1 provides a comparative overview of traditional MoE and our proposed UoE architecture. The first key advancement of UoE is to ***adopt only a small partial of***

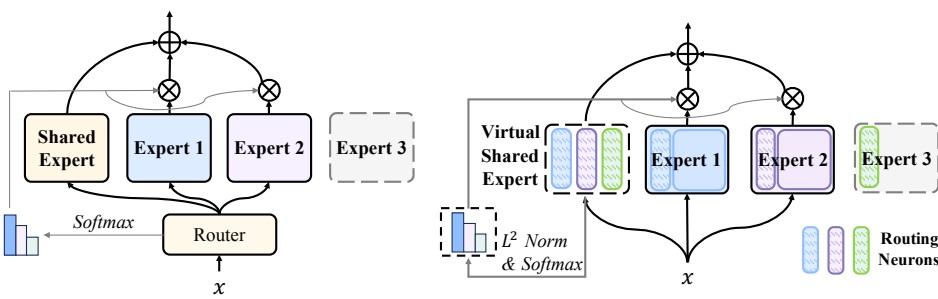

(a) Mixture-of-Experts          (b) Union-of-Experts

Figure 1: A comparison of Mixture-of-Experts (MoE) and Union-of-Experts (UoE) routing mechanisms. In MoE, Expert 1 & 2 are selected based on highest router logits. In UoE, Expert 1 & 2 are selected as its routing neurons exhibit the largest activation norms. Gray modules are inactive; regions with diagonal stripe denote the routing neurons within the weight matrix.

*expert neurons to represent the activation degree of an entire expert*, based on a surprising finding: only a small subset of $N_s \ll D$ neurons within each expert weight, referred to *routing neurons*, is sufficient to parameterize the routing function, where $D$ is the dimension of the intermediate activations. This reduces the computational overhead of AoE to a fraction of $N_s/D$. Our analysis show that the selection of routing neurons can be highly flexible. By simply pre-designating the first $N_s$ neurons in expert's weight matrix as routing neurons before training, their activations spontaneously exhibit high correlation with those of the entire weight matrix. This indicates that these neurons can effectively represent the behavior of the majority of neurons within the expert.

Nevertheless, computing the routing neurons in each expert still introduces additional overhead. To eliminate this remaining cost, the second key advancement of UoE is to *pack routing neurons from each expert into a virtual shared expert*. This approach is grounded in a key insight: the shared expert (Dai et al., 2024) widely used in MoEs, which processes all tokens to consolidate common capabilities implicitly scattered across individual experts. UoE explicitly implements this common capability consolidation by reusing the already-computed routing neurons, which perform the common routing function, to collectively form the output of this "virtual" shared expert. By "virtual," we mean that this is not a materialized module but a conceptual structure, describing how the outputs of routing neurons—which remain within their original experts—are reused collectively. Consequently, the computational cost of these neurons is reused rather than wasted. This allows UoE to achieve computational and memory costs identical to a standard MoE architecture while delivering superior performance.

We pre-train UoE with up to 3 billion parameters, achieving superior performance over both MoE and AoE while keeping the inference cost on par with MoE. Additionally, we present a thorough model analysis of UoE to underscore its advantages, such as improved load balance.

## 2   BACKGROUND AND NOTATION

### 2.1   MIXTURE-OF-EXPERTS (MOE)

We adopt the Gated Linear Unit (GLU) as the expert module, following mainstream MoE designs (Dai et al., 2024; Jiang et al., 2024). The $i$-th expert is parameterized by three matrices: $\mathbf{W}_g^i, \mathbf{W}_p^i \in \mathbb{R}^{d \times D}$ and $\mathbf{W}_o^i \in \mathbb{R}^{D \times d}$, with its forward pass defined as:

$$\mathrm{E}_i(\boldsymbol{x}) = \left(\texttt{SiLU}(\boldsymbol{x}\boldsymbol{W}_g^i) \odot (\boldsymbol{x}\boldsymbol{W}_p^i)\right)\boldsymbol{W}_o^i. \tag{1}$$

An MoE FFN layer consists of $N$ experts, with $K$ experts selected to process an input token $\boldsymbol{x}$. Adopting the design from (Dai et al., 2024), we also include a shared expert $\mathrm{E}_s$ that processes all tokens. This shared expert captures the common capabilities, allowing the other experts to become more specialized.

The output of an MoE FFN layer is the sum of two components: the output of a shared expert and a weighted sum of the selected expert outputs. The weights for the latter are given by a router parameterized by a matrix $\boldsymbol{R} \in \mathbb{R}^{d \times N}$:

$$G(\boldsymbol{x}) = \text{softmax}(\boldsymbol{x}\boldsymbol{R}),$$
$$\text{FFN}(\boldsymbol{x}) = \text{E}_s(\boldsymbol{x}) + \sum_{i \in \text{TopK}(G(\boldsymbol{x}))} G(\boldsymbol{x})[i] \cdot \text{E}_i(\boldsymbol{x}). \tag{2}$$

## 2.2 Autonomy-of-Experts (AoE)

AoE (Lv et al., 2025) addresses the misalignment between router decisions and experts' actual capabilities by encoding the routing function $G(\boldsymbol{x})$ into the expert parameters themselves. The key insight is that the intermediate activation magnitude of an expert indicates how well its capabilities match the input token's requirements.

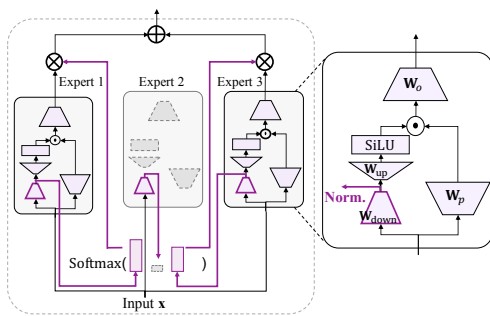

Figure 2: The overview of an AoE model.

To reduce the computational cost associated with $D$-dimensional activations, $\boldsymbol{W}_g^i$ is replaced with two low-rank matrices: $\boldsymbol{W}_{down}^i \in \mathbb{R}^{d \times r}$ and $\boldsymbol{W}_{up}^i \in \mathbb{R}^{r \times D'}$. The intermediate dimension $D'$ is chosen to preserve the same number of parameters as the original MoE, and is given by:

$$D' = \frac{3Dd - dr}{r + 2d}.$$

Each token is multiplied by all $\boldsymbol{W}_{down}^i$ matrices, and the L$^2$-norms of the resulting $N$ activations (each of dimension $r$) are used for expert selection. Experts with the top-$K$ activation norms continue forward computation, while unselected experts terminate early. The routing function $G$ and the forward pass for selected experts are defined as:

$$G(\boldsymbol{x}) = \text{softmax}([g_1, g_2, \cdots, g_n]), \text{ where } g_i = \|\boldsymbol{x}\boldsymbol{W}_{down}^i\|,$$
$$E_i(\boldsymbol{x}) = \left(\text{SiLU}(\boldsymbol{x}\boldsymbol{W}_{down}^i \boldsymbol{W}_{up}^i) \odot (\boldsymbol{x}\boldsymbol{W}_p^i)\right) \boldsymbol{W}_o^i. \tag{3}$$

While AoE's autonomous expert selection leads to better downstream task performance than MoE, it introduces computational and memory overhead. The inefficiency arises because all experts compute activations, but only a fraction are used in the output. This waste scales with an increased $N$ and a decreased $K$. Therefore, this paper focuses on achieving autonomous selection with an efficiency comparable to vanilla MoE, independent of $N$ and $K$.

## 3 Methodology

### 3.1 Motivation

To improve efficiency, AoE introduces factorization of $\boldsymbol{W}_g$. Paradoxically, this design traps AoE in a dilemma: it must contend with either substantial computational overhead or excessive memory access. Consequently, factorization itself becomes the fundamental bottleneck to further efficiency-wise advancement in AoE. Our following analysis reveals this inherent dilemma. The detailed derivation of the results in this subsection can be found in Appendix A.

We show that, in theory, AoE introduces additional FLOPs per token, which grow linearly with the factorization rank $r$ compared to a vanilla MoE (with identical parameter count) as:

$$\Delta\text{FLOPs} = 2 \cdot d \cdot r \cdot (N - K). \tag{4}$$

Additionally, AoE incurs extra memory overhead (per token) given by:

$$\Delta\text{Mem} = \max(Nr, \ 4K(D' - D)). \tag{5}$$

We visualize AoE's computational and memory overhead as a function of $r$ in Figure 3. The results clearly show that regardless of the value of $r$, AoE is bounded by either memory or computational resources. A rank $r$ between 48 and 80 offers a relatively more favorable trade-off: although it still incurs significant memory overhead, the computational cost is substantially reduced. However, for wide models with large $d$ and $D$, setting $r$ this low leads to unstable training of AoE, rendering this theoretically optimal range impractical.

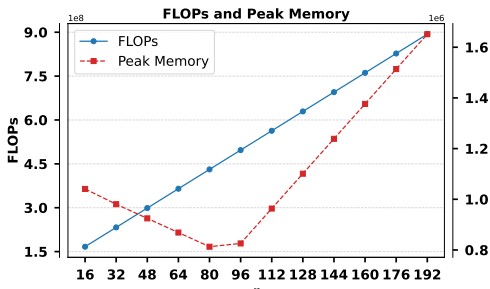

Figure 3: The efficiency dilemma introduced by factorization: for any rank $r$, AoE is theoretically bounded by either computation or memory.

This dilemma motivates a new realization of autonomous expert selection, which for practicality and scalability must improve efficiency by eliminating the root cause of waste rather than relying on low-rank factorization.

### 3.2 Routing neurons accelerate autonomous routing

**Model structure** Through extensive trials, we identified a promising approach that successfully maintains autonomous expert selection based on activation norms while achieving high efficiency without relying on factorization. As no factorization is applied, each expert in our model, namely UoE, is parameterized identically to a vanilla MoE (Eq. 1) using standard dense weight matrices.

We find that only a small subset of neurons within each expert's weight matrix is sufficient to parameterize the routing function. We refer to these as *routing neurons*. Notably, the selection of these routing neurons proves highly flexible (refer to Appendix B for more details). UoE operates by simply pre-designating the first $N_s \ll D$ neurons of each expert weight matrix as routing neurons before training. These neurons, being part of an expert's parameters, are marked with a tilde superscript:

$$\widetilde{\boldsymbol{W}_g^i} = \boldsymbol{W}_g^i[:,:N_s], \quad \widetilde{\boldsymbol{W}_p^i} = \boldsymbol{W}_p^i[:,:N_s], \quad \widetilde{\boldsymbol{W}_o^i} = \boldsymbol{W}_o^i[:N_s,:],$$

For any input $x$, UoE performs autonomous expert selection based on the activation intensity (measured by $L^2$ norm) of routing neurons. This approach is motivated by prior work (Lv et al., 2025; Geva et al., 2021) which establishes that high activation magnitude indicates a module is well-aligned with the input. Another fundamental premise of UoE is that the activation magnitude of the routing neurons is highly correlated with that of their entire expert, a correlation we show in Section 4.4 is spontaneously reinforced during training.

Formally, we define the routing function $G$ in UoE as:

$$\mathrm{G}(\boldsymbol{x}) = \mathrm{softmax}\left(\mathrm{TopK}\left[g_1, g_2, \cdots, g_n\right]\right), \text{ where}$$
$$g_i = \|\mathtt{SiLU}(\boldsymbol{x}\widetilde{\boldsymbol{W}_g^i}) \odot (\boldsymbol{x}\widetilde{\boldsymbol{W}_p^i})\|. \tag{6}$$

Because these routing neurons separately located in each routed expert collaboratively function as an "autonomous routing function", UoE, like AoE, eliminates the separate, explicit router module.

### 3.3 Virtual shared expert improves activation utilization efficiency

We observe that routing neurons, activated on every token, functionally resemble a shared expert, which processes all tokens regardless of which experts are selected or not. We therefore consolidate them into a virtual shared expert, which replaces the conventional shared expert. This ensures the contributions of routing neurons from unselected experts are not wasted, fundamentally resolving the inherent computation and memory inefficiencies of AoE models. By "virtual," we mean that during training, these neurons are not physically restructured into a single module but remain within their original experts; their consolidation is an abstract concept describing how their activations are collectively reused beyond mere routing.

To be specific, the virtual shared expert consists of three virtual matrices during training:

$$W_g^s = \left(\ \widetilde{\boldsymbol{W}_g^1}\ \middle|\ \widetilde{\boldsymbol{W}_g^2}\ \middle|\ \dots\ \middle|\ \widetilde{\boldsymbol{W}_g^N}\ \right),$$

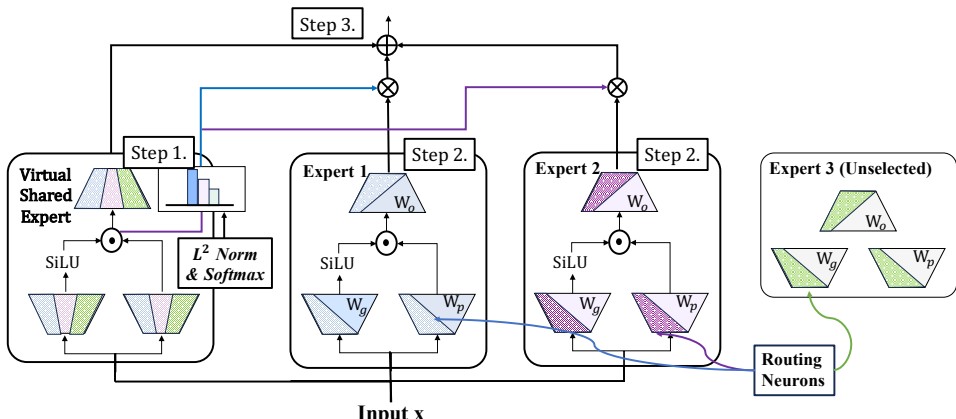

Figure 4: In UoE, the first $N_s$ neurons in each parameter matrix are designated as routing neurons. These neurons process every token, with their activations used to compute routing logits. During training, while these neurons remain distributed across experts, they collectively function as a virtual shared expert—their outputs contribute to the final prediction like a standard shared expert, regardless of whether their host expert is selected. During inference, this virtual expert is materialized as a single module. The forward pass of UoE consists of three steps: (1) computing the activations of the routing neurons to obtain routing logits (also obtaining the output of the virtual shared expert), (2) performing expert routing using the routing logits and activating selected routed experts, and (3) merging the outputs of the virtual shared expert with those of the routed experts.

$$W_p^s = \left(\ \widetilde{\boldsymbol{W}_p^1} \ \Big| \ \widetilde{\boldsymbol{W}_p^2} \ \Big| \ \dots \ \Big| \ \widetilde{\boldsymbol{W}_p^N} \ \right),$$

$$W_o^s = \left(\ \widetilde{\boldsymbol{W}_o^1}^\top \ \Big| \ \widetilde{\boldsymbol{W}_o^2}^\top \ \Big| \ \dots \ \Big| \ \widetilde{\boldsymbol{W}_o^N}^\top \ \right)^\top.$$

We set the number of routing neurons per parameter matrix as $N_s = \text{round}(D/K)$, matching the parameter count of a standard shared expert. This ensures that UoE has identical memory and computational overhead to a conventional MoE with the same $N$ and $K$.

During inference, the virtual shared expert is materialized as a single module, ensuring UoE's checkpoint compatibility with all well-developed kernels designed for accelerating standard MoE models. A detailed implementation for UoE's training and inference is provided in Appendix C.3.

## 4 EXPERIMENTS

### 4.1 MAIN RESULTS AND ANALYSIS

**General Setup.** We pre-train language models with 1B parameters to verify the effectiveness of UoE. Our language model consists of 8 Transformer layers. For each Transformer layer, we employ the multi-head attention mechanism with a total of 8 attention heads. We substitute all FNN layers with MoE layers while keeping the number of expert activations consistent across all methods. The MoE baseline is configured with a shared expert following the setup in (Dai et al., 2024). Due to the page limit, we present more details about our architecture and implementations in Appendix C.

We pre-train our language models with 100B tokens from FineWeb datasets (Penedo et al., 2024), and use the Llama tokenizer for tokenization. For training setups, we employ the AdamW optimizer with $(\beta_1, \beta_2) = (0.9, 0.95)$, a gradient norm clipping threshold of 1, and weight decay as 0.1. We use a learning rate of $1 \times 10^{-3}$ with 1000 steps linear warmup, followed by a cosine decay scheduler.

We evaluate these language models across 8 widely used benchmarks, including *ARC* (Clark et al., 2018), *PIQA* (Bisk et al., 2020), *HellaSwag* (Zellers et al., 2019), *SCIQ* (Welbl et al., 2017), *Winogrande* (Sakaguchi et al., 2019), *MNLI* (Wang et al., 2018), *QNLI* (Wang et al., 2018) and *RTE* (Wang et al., 2018). These benchmarks assess the models' capabilities in language understanding, question answering, and natural language inference. All evaluations are performed using the LM Evaluation

Table 1: Results for the validation experiments on 1B parameter language models. We compare models with different numbers of activated experts, both with and without the auxiliary load balancing loss. Colored entries highlight improvements over the MoE baseline, while bold text mark the best results within each experimental setting.

| Model | Num. | $\mathcal{L}_{aux}$ | ARC-E | PIQA | HELLA | SCIQ | WINO | MNLI | QNLI | RTE | AVG. |
|---|---|---|---|---|---|---|---|---|---|---|---|
| MoE | 8 | ✓ | 62.54 | 68.88 | 36.74 | 81.60 | 52.49 | 32.78 | **51.04** | 49.46 | 54.44 |
| AoE | 8 | ✓ | **64.60** | 69.59 | 36.62 | **83.30** | 51.22 | **34.13** | 50.01 | 48.86 | 54.79 |
| UoE | 8 | ✓ | 63.09 | **69.64** | **37.07** | 82.40 | **52.88** | 33.89 | 50.05 | **51.50** | **55.07** |
| MoE | 8 | ✗ | 62.75 | 68.23 | 36.62 | 81.10 | 51.85 | 33.12 | **49.95** | 50.18 | 54.23 |
| AoE | 8 | ✗ | 62.29 | 68.17 | 36.32 | **82.20** | **54.14** | **33.71** | 49.78 | 49.10 | 54.46 |
| UoE | 8 | ✗ | **64.56** | **69.10** | **36.86** | 81.50 | 52.09 | 33.02 | 49.91 | 49.46 | **54.56** |
| MoE | 4 | ✓ | 61.45 | 67.52 | 35.27 | 77.10 | 50.75 | 33.25 | 49.83 | 46.45 | 52.70 |
| AoE | 4 | ✓ | 61.57 | 68.61 | 36.07 | **82.40** | 52.01 | 33.12 | 49.80 | **50.30** | 54.24 |
| UoE | 4 | ✓ | **62.25** | **68.66** | **35.67** | 81.70 | **54.70** | **33.62** | 50.20 | 48.98 | **54.47** |

Harness (Gao et al., 2024). The first five tasks are evaluated zero-shot. For the remaining three tasks, we report their average performance under 0-shot, 3-shot and 5-shot to reduce randomness.

**Experimental Results.** We present the main results in Table 1. We pre-train 1B-parameter language models with varying number of expert activation, both with and without the auxiliary load-balancing loss. UoE consistently outperforms both MoE and AoE models in overall performance across all of these configurations, which further demonstrates the effectiveness of UoE's model design (note that UoE is more efficient than AoE, refer to Section 4.2 for more detailed discussions).

Notably, UoE achieves more substantial performance improvements under a sparser expert activation setting (activating 3 out of 64 experts), which is a defining characteristic of modern MoE architectures. It implies that UoE could better select effective expert combinations among larger numbers of routed experts. Figure 5 illustrates the pre-training negative negative log-likelihood (NLL) loss of UoE and baseline methods in this setup. UoE exhibits a lower training loss during the pre-training phase, indicating its higher efficiency in parameter updates.

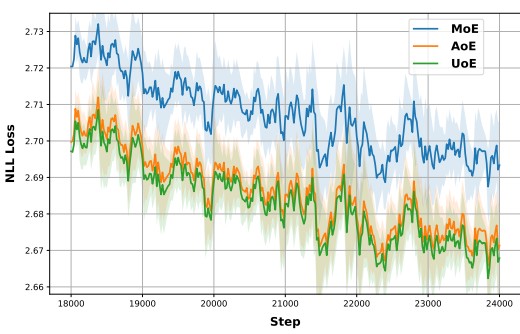

Figure 5: Pre-training NLL loss comparison.

### 4.2 EFFICIENCY ANALYSIS OF UoE

In this section, we analyze the efficiency of UoE in comparison with the baseline methods, focusing primarily on (1) training efficiency metrics and (2) expert loading balance.

**TFLOPS, Peak Memory and Throughput.** We begin by conducting a comparative analysis of UoE's training efficiency. Table 2 reports the training achieved TFLOPS, peak memory usage and throughput of UoE and baseline methods during pre-training.

Table 2: Achieved training TFLOPS, Memory and Throughput.

| | TFLOPS | Mem.(GB) | TP. (K/s) |
|---|---|---|---|
| MoE | 90.40 | 63.93 | 604.00 |
| AoE | 78.29 | 71.51 | 509.00 |
| UoE | 86.51 | 63.96 | 610.00 |

We observe that UoE achieves a 19.8% improvement in training throughput over AoE while maintaining downstream performance that is better or competitive with AoE, and superior to MoE. Meanwhile, UoE incurs computational overhead that is nearly identical to MoE at inference time. Consequently, we contend that UoE is an

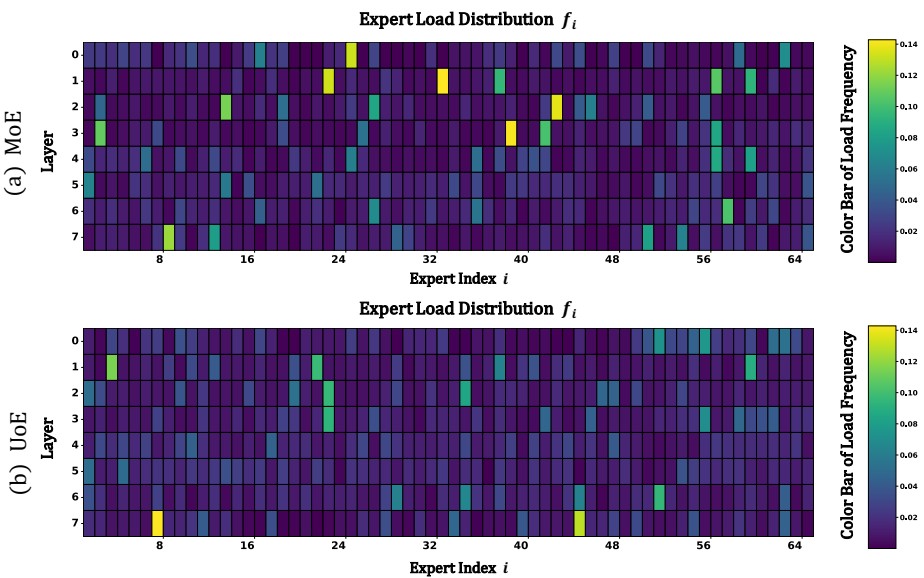

Figure 6: Expert Loading Distribution of UoE and MoE.

efficient implementation for expert autonomy and resolve the dilemma of AoE without compromising on effectiveness.

**Load Balance of UoE.**   The imbalanced expert load is a critical challenge leading to the computational overhead of MoEs (Fedus et al., 2022). Prior study shows that AoE achieves better load balancing than traditional MoE.

We compare UoE with MoE to investigate whether UoE can enhance load balancing in the absence of an auxiliary loss. Specifically, we sample 1,000 instances from Wikitext-2 (Merity et al., 2016) as a calibration set and examine their expert loading patterns. Figure 6 visualizes the expert loading for our pre-trained UoE alongside MoE, where the load distribution $f_i$ for the $i$-th expert on a batch of $T$ tokens is defined as:

$$\mathbf{f}_i = \frac{1}{T} \sum_{\mathbf{x} \in \mathcal{B}} \mathbb{1} \left\{ i \in \texttt{argtopK}\left(G\left(\mathbf{x}\right)\right) \right\}.$$

Except for the final layer, UoE achieves consistently better load balance, with far fewer cases of the imbalance observed in the shallow layers of MoE. Table 3 compares layer-wise entropy of expert selection of MoE and UoE to highlight their differences. The results shows that UoE maintains a more balanced expert load across nearly all layers, even without an auxiliary load-balancing loss.

Table 3: Entropy of expert selection. Higher entropy indicates more balanced expert loads.

|  | $\text{Ent}_{\text{load}}^1$ | $\text{Ent}_{\text{load}}^2$ | $\text{Ent}_{\text{load}}^3$ | $\text{Ent}_{\text{load}}^4$ | $\text{Ent}_{\text{load}}^5$ | $\text{Ent}_{\text{load}}^6$ | $\text{Ent}_{\text{load}}^7$ | $\text{Ent}_{\text{load}}^8$ |
|---|---|---|---|---|---|---|---|---|
| **MoE** | 3.45 | 3.23 | 3.29 | 3.14 | 3.57 | 3.76 | **3.66** | **3.42** |
| **UoE** | **3.70** | **3.62** | **3.71** | **3.71** | **3.88** | **3.84** | **3.66** | 3.31 |

## 4.3   ABLATION STUDIES

We take our pre-trained UoE with the auxiliary balancing loss, keeping 8 experts activated as the basic setup, to evaluate the effectiveness of UoE's various designs.

**Ablation Study of the Virtual Shared Expert.**   We perform ablation experiments to valid its contribution. We first highlight that the virtual shared expert is crucial in pre-trained UoE models. For configurations ① and ②, we deactivate different experts in the pre-trained language model and

Table 4: Analysis of model designs in UoE through ablation studies.

| | Configuration | ARC-E | PIQA | HELLA | SCIQ | WINO | MNLI | QNLI | RTE | AVG. |
|---|---|---|---|---|---|---|---|---|---|---|
| | **UoE** | 63.09 | 69.64 | 37.07 | 82.40 | 52.88 | 33.89 | 50.05 | 51.50 | **55.07** |
| ① | *w.o* shared expert-v1 | 53.83 | 66.21 | 33.80 | 75.80 | 50.36 | 33.93 | 50.27 | 51.50 | 51.96 |
| ② | *w.o* shared expert-v2 | 62.42 | 69.48 | 37.16 | 81.90 | 52.09 | 33.76 | 49.86 | 51.74 | 54.80 |
| ③ | *w.o* shared expert-v3 | 65.19 | 69.53 | 36.67 | 81.60 | 49.88 | 33.52 | 50.05 | 49.22 | 54.46 |
| ④ | double $N_s$ | 63.72 | 68.28 | 36.58 | 84.20 | 51.30 | 34.11 | 49.97 | 50.66 | 54.85 |
| ⑤ | $\mathbf{x}\mathbf{W}_p$ | 63.72 | 70.08 | 36.69 | 82.50 | 51.70 | 32.95 | 50.00 | 50.18 | 54.73 |
| ⑥ | $\mathbf{x}\mathbf{W}_g$ | 63.97 | 69.21 | 37.25 | 80.70 | 52.09 | 33.62 | 51.06 | 49.58 | 54.69 |
| ⑦ | $\texttt{SiLU}(\mathbf{x}\mathbf{W}_g)$ | 63.51 | 69.48 | 36.76 | 82.40 | 53.35 | 33.73 | 49.58 | 49.22 | 54.75 |

observe its downstream performance changes. In configuration ①, we disable the virtual shared expert and activate only the routed experts; In configuration ②, we always keep the virtual shared expert active and reduce the number of activated experts to ensure a fair comparison. Given this, we find that the shared expert exerts a significant impact on downstream performance. Configuration ① demonstrates substantially inferior performance compared with ②. This indicates that the shared expert in UoE truly learns abilities compulsory that the routed experts have not captured.

We also pre-train UoE without the virtual shared expert from scratch. In configuration ③, the routing neurons are not reactivated and are used simply for expert routing. As shown in Table 4, the absence of the activated shared expert once again leads to a decline in model performance.

**Ablations Study on the Selection of $N_s$.** We perform ablation studies to investigate the effect of varying $N_s$, the number of routing neurons. Specifically, we double the number of routing neurons and pre-train the model from scratch. This setting will increases an extra shared experts, while the number of activated routed experts is reduced to keep the total count of active experts constant. We do not explore alternative settings, as they would result in an excessive number of shared experts. Our results show that even doubling the number of routing neurons does not improve model performance and may even cause a slight degradation in capability.

**Ablation Study of Expert Selection Strategies.** By default, we use the activation intensity of the $i$-th expert for routing decisions, which is formulated in Equation 6. For configurations ⑤ to ⑦, we attempt to use the $L^2$ norm of other intermediate nodes within the computation graph for routing.

We pre-train these variants from scratch and present their downstream performance in Table 4, together with the nodes used for norm calculation. The results show that these variants achieve overall performance comparable to the default configuration ($\texttt{SiLU}(\mathbf{x}\mathbf{W}_g) \odot \mathbf{x}\mathbf{W}_p$), albeit slightly lower.. Their training time is nearly identical. Overall, these results justify the use of activation intensity.

### 4.4 CONSISTENCY OF EXPERT SELECTION BETWEEN EXPERT AND ROUTING NEURON ACTIVATIONS

We argue that the activation patterns of routing neurons closely reflect those of their corresponding experts. To clarify that, we perform expert routing based on the experts' activation intensity, rather than the routing neurons' in a pre-trained UoE model. To be specific, we activate the top-k experts with the highest activation values, and directly evaluate UoE's downstream performance without further training. Table 5 presents the results, and only a minor performance drop is observed.

Table 5: Performance change when using experts' activation intensity instead of routing neurons.

| Model | Act. | ARC-E | PIQA | HELLA | SCIQ | WINO | MNLI | QNLI | RTE | AVG. |
|---|---|---|---|---|---|---|---|---|---|---|
| UoE | Neurons | 63.09 | 69.64 | 37.07 | 82.40 | 52.88 | 33.89 | 50.05 | 51.50 | 55.07 |
| | Expert | 61.49 | 68.72 | 36.49 | 82.10 | 51.54 | 34.25 | 50.16 | 50.30 | 54.38 |

Table 6: For 3B-paramter LLMs, UoE exhibits consistent downstream performance. Colored entries show improvements over the MoE baseline; bold text indicates the best results.

| Model | ARC-E | ARC-C | PIQA | HELLA | SCIQ | WINO | AVG. |
|-------|-------|-------|------|-------|------|------|------|
| MoE | 63.64 | 31.48 | 70.62 | 39.52 | **89.40** | 51.22 | 57.65 |
| AoE | 64.44 | 31.57 | 70.24 | 40.34 | 88.80 | **53.35** | 58.12 |
| UoE | **69.07** | **33.11** | **73.18** | **41.96** | 87.10 | 52.80 | **59.54** |

## 4.5 VALIDATION OF UoE WITH LARGER MODEL SIZE

We pre-train UoE and its competitors with a total of 3 billion parameters. We follow most of the architectural settings from Section 4.1. For these 3B-parameter language models, each model consists of 20 layers and 20 attention heads, with the hidden dimension expanded to 1280. The number of experts is kept consistent with the previous setup, and 7 routed experts are activated. We adjust training parameters accordingly to better suit the training.

At larger parameter scales, UoE consistently outperforms MoE and AoE models, with improvements becoming increasingly pronounced as the model size grows. This highlights the potential of scaling UoE to even greater parameter sizes to further boost its capabilities.

## 5 RELATED WORK

**Mixture-of-Experts.** The Mixture-of-Experts (MoE) paradigm was originally proposed as a modular neural network framework in which a gating function assigns inputs to specialized experts (Jacobs et al., 1991; Jordan & Jacobs, 1994). More recently, MoE has been integrated into large-scale Transformers to achieve trillion-parameter models with sparse computation (Lepikhin et al.; Fedus et al., 2022). Subsequent work has focused on improving efficiency through balanced expert assignment (Lewis et al., 2021) and system-level optimizations for distributed training (Hwang et al., 2023; Gale et al., 2022). Despite these advances, sparse MoE models continue to face challenges such as routing instability and expert redundancy. To mitigate these issues, DeepSeekMoE (Dai et al., 2024) introduces shared experts, which provide stable coverage of common knowledge while routed experts focus on specialization. In addition, its fine-grained expert partitioning further enhances efficiency and encourages more diverse expert behaviors. In this work, we adopt most of the configurations from DeepseekMoE. In contrast, our virtual shared expert is constructed from all routing neurons, thereby functioning both as the shared expert and as the mechanism for autonomous routing.

**Expert Selection Strategies.** Prior work on expert selection has explored a variety of routing mechanisms to determine which experts to activate from a set of $N$ candidates. Top-k routing (Lepikhin et al.) activates a fixed number of experts per token based on router-assigned scores, while Top-$p$ routing dynamically selects experts until a cumulative probability threshold $p$ is reached. Despite these differences, most approaches rely on a centralized router to assign tokens to experts. In contrast, Lv et al. (2025) eliminates the router entirely by allowing experts to self-activate, thereby achieving expert selection in a fully decentralized manner. In this paper, we improve AoE's expert autonomy by addressing efficiency issues and replacing low-rank factorization with routing neurons.

## 6 CONCLUSION

In this paper, we introduce UoE, a novel MoE variant that perform expert autonomy routing. UoE leverages only a small subset of neurons in each expert to capture the expert's overall activation, effectively addressing the efficiency challenges encountered in previous work. Moreover, we treat these routing neurons collectively as a shared expert to further enhance activation utilization efficiency. We hope that UoE can inspire the community to pursue more effective autonomy-based routing strategies to mitigate the decoupling between routing decisions and expert capabilities.

ETHICS STATEMENT

This work focuses on the development of a Mixture-of-Experts (MoE) model. Our study does not involve human subjects, personally identifiable information, or sensitive data. We do not foresee any direct ethical or societal risks arising from our methodology or experiments.

REPRODUCIBILITY STATEMENT

We have added our code to the supplementary materials, and all the data used is open-source. The experimental setup is detailed in Section 4.1. Unless noted, all experiments use the same settings. Overall, these practices make our results reproducible.

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

## A    DISCUSSION OF THE TRADE-OFF BETWEEN MEMORY CONSUMPTION AND COMPUTATIONAL OVERHEAD

For simplification, we omit the cost of the router and the FLOPs of a traditional MoE layer is:

$$\text{FLOPs} = 3 \cdot \text{TK} \left( 2\,\text{D} \cdot \text{d} \right).$$

For an arbitrary AoE layer, the FLOPS it requires is:

$$\text{FLOPs} = 2 \cdot \text{TK} \left( 2\,\text{D}' \cdot \text{d} \right) + \text{TK} \left( 2\,\text{D}' \cdot \text{r} \right) + \text{TN} \left( 2\,\text{d} \cdot \text{r} \right),$$

where $\text{D}'$ is the FFN hidden size of AoE to ensure the same number of parameters as MoE as:

$$D' = \frac{3 \cdot \text{D} \cdot \text{d} - \text{d} \cdot \text{r}}{\text{r} + 2 \cdot \text{d}}.$$

Compared with MoE, AoE introduces an overhead of FLOPs that is:

$$\Delta\text{FLOPs} = 2\,\text{T} \cdot \text{d} \cdot \text{r} \cdot (N - \text{K}). \tag{7}$$

## B    TRIALS ON SELECTING ROUTING NEURONS WITHIN EXPERTS

Motivated by our preliminary explorations, we investigate the idea of fixing a subset of neurons as routing neurons to enable expert autonomy. In FFNs, neurons are dynamically activated based on input. Despite that, our goal is to identify a subset of key neurons that effectively capture the overall activation pattern. Our initial approach dynamically selects important neurons during training and then fixes this subset during inference, allowing dominant weights in the experts' parameters to be located on the fly. A simple strategy uses the $L^2$-norm to identify high-contributing neurons. Although this incurs higher training cost than standard MoE, it remains substantially more efficient than AoE. To further improve efficiency, we explore whether permanently fixing neurons could work. Our experiments further confirm its feasibility.

## C    IMPLEMENTATION DETAILS OF UoE

### C.1    HYPER-PARAMETERS OF MODEL ARCHITECTURE

Table 7 presents details on the architecture hyper-parameters used throughout our experiments.

### C.2    TRAINING SETUP DETAILS FOR UoE

We provide additional details on our efficient training of UoE. The training pipeline is built upon TorchTitan framework (Liang et al., 2025), uses PyTorch's scaled_dot_product_attention for attention, and adopts the MegaBlocks (Gale et al., 2022) MLP for MoE layer implementation.

Table 7: Hyper-parameters of model architecture.

| Hyper-Parameters | 1B | 3B |
|---|---|---|
| hidden size | 1024 | 1280 |
| MoE layers | 8 | 20 |
| FFN hidden size | 512 | 512 |
| attention heads | 8 | 8 |
| key-value heads | 20 | 20 |
| routed experts | 64 | 64 |
| vocab size | 128,256 | 128,256 |
| RoPE theta | 500,000 | 500,000 |

### C.3    IMPLEMENTATION DETAILS OF THE VIRTUAL SHARED EXPERT

Figure 10 presents a naive PyTorch implementation of UoE's training and inference. The slight difference lies in repacking the routing neurons, originally distributed across different experts, into a layout conforming to the MoE shared expert. This prevents non-contiguous parameter access at inference time and improves UoE's compatibility with practical MoE deployments, such as Expert Parallelism. More details can be found in our code repository.

# D TOWARD A MECHANISTIC UNDERSTANDING OF ROUTING NEURONS

In this section, we aim to provide a theoretical explanation for how routing neurons can reflect an expert's activation. Specifically, an expert's activation is jointly determined by how the input $x$ activates with both $W_g$ and $W_p$. Without loss of generality, we take $xW_g$ as the running example in the discussion below. Following Lv et al. (2025), we measure the activation intensity of the input token $x$ at $W_g$ via the $\mathtt{L}^2\text{-Norm}$ of $xW_g$, which can be formulated as:

$$\mathtt{L}^2\text{-Norm}\left(xW_g\right) = \sqrt{xW_g W_g^\top x^\top}.$$

Given the singular value decomposition of $W_g$, we can expand this equation into:

$$\mathtt{L}^2\text{-Norm}\left(xW_g\right) = \sqrt{xU_g\Sigma_g^{\,2}U_g^\top x^\top}, \quad \text{where } W_g = U_g\,\Sigma_g\,V_g^\top.$$

Similarly, the activation intensity of input token $x$ at $\widetilde{W_g}$ is given by:

$$\mathtt{L}^2\text{-Norm}\left(x\widetilde{W_g}\right) = \sqrt{xU_r\Sigma_r^{\,2}U_r^\top x^\top}, \quad \text{where } \widetilde{W_g} = U_r\,\Sigma_r\,V_r^\top.$$

As the expert weights of MoE models are intrinsically low-rank (Lv et al., 2025; Gu et al., 2025), the $\mathtt{L}^2\text{-Norm}(xW_g)$ is dominated by a small portion of the singular vectors with the largest singular values. Considering that, we compute and plot the similarity between $U_r$ and the principal singular vectors of $W_g$ across all experts in UoE with 1B parameters.[1]

Figure 7 visualizes the results, where $S_g^{i,j}$ of the $j$-th expert at layer $i$ is defined as:

$$S_g^{i,j} = \mathtt{L}^2\text{-Norm}\left(< U_r, U_g[0] >\right),$$

we also visualize the similarity between the router weights $R^{i,j}$ in the MoE baseline and the principal singular vectors for comparison. $S_r^{i,j}$ is denoted as:

$$S_r^{i,j} = < \frac{R^{i,j}}{||R^{i,j}||},\ U_g[0] >,$$

where $R^{i,j}$ is the $i$-th row of the router weights at layer $j$.

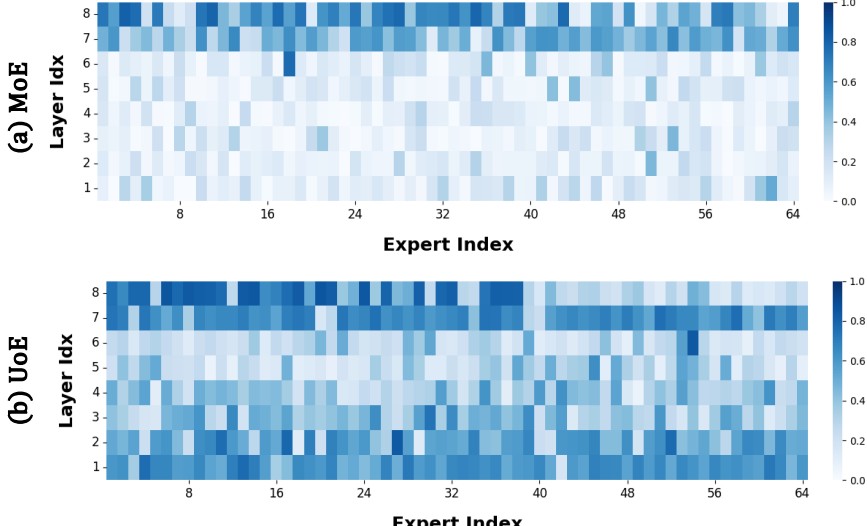

Figure 7: Heatmap visualization of $S_g^{i,j}$ and $S_r^{i,j}$ across experts and layers.

---

[1]We use $U_g[0]$ to denote the principal singular vector with the largest singular value.

As shown in Figure 7, the principal singular vector of these routing neurons exhibits a noticeable similarity to that of the expert weight matrices, whereas no similar phenomenon was observed in MoE. We argue that this behavior arises from the specialized training dynamics of expert autonomy. This alignment serves as the underlying mechanism that enables them to represent expert activations.

A similar pattern also emerges in $W_p$, and we provide the corresponding visualization in Figure 8.

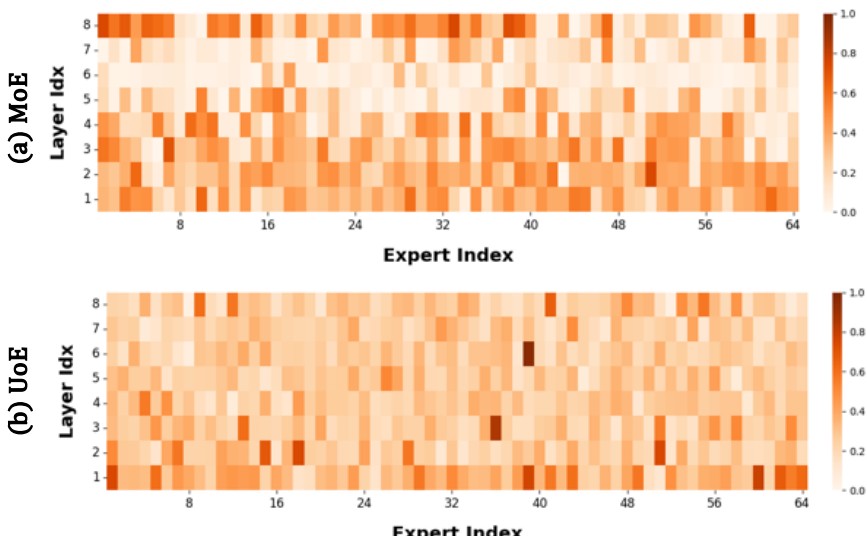

Figure 8: Heatmap visualization of $S_p^{i,j}$ and $S_r^{i,j}$ across experts and layers.

# E    EFFICIENCY ANALYSIS FOR UoE AT INFERENCE TIME

To evaluate the inference performance of UoE, we build a generation pipeline on top of HuggingFace's GenerationMixin (Wolf et al., 2020). We conduct a breakdown analysis of UoE's inference efficiency, benchmarking the peak memory occupation and end-to-end generation throughput. We use 256 random tokens as input and conduct experiments across different generation lengths and batch-size configurations.

Table 8 presents the results; we can conclude that the computation overhead of UoE is nearly identical to that of MoE. More implementation details can be found in Section C.3.

Table 8: Throughput and peak memory usage comparisons.

| Configuration | | TP. (token/s) / Mem. (GB) | | |
|---|---|---|---|---|
| Model | BS | 256 | 1024 | 4096 |
| MoE | 1 | 35.71 (2.15) | 35.84 (2.15) | 35.81 (2.15) |
| UoE | | 35.84 (2.15) | 35.98 (2.15) | 35.89 (2.15) |
| MoE | 4 | 141.99 (2.21) | 141.38 (2.21) | 141.37 (2.21) |
| UoE | | 140.83 (2.21) | 141.00 (2.21) | 141.46 (2.21) |
| MoE | 16 | 561.82 (2.48) | 560.93 (2.48) | 559.97 ( 2.48) |
| UoE | | 549.91 (2.47) | 551.39 (2.47) | 551.93 (2.47) |

## F    VIRTUAL SHARED EXPERT IN UOE IS ALSO A COMMON-KNOWLEDGE CONSOLIDATOR

Given that the routing neurons are always activated in UoE's forward pass, we reuse the intermediate hidden states and introduce the virtual shared expert. In this section, we show that this design not only improves activation reuse and reduces overhead, but also facilitates knowledge sharing.

Geva et al. (2021) interpret transformer FFN layers as key-value memories, with knowledge or abilities stored in the "value" vector (i.e., $W_o$ in Gated Linear Unit). Following this intuition, we perform PCA to project each row of $W_o$ from all experts into a 2D space. We visualize the projected expert weights in UoE and observe that the routing neurons concentrate primarily along the leading principal component, with notably large projections onto this direction. This hints that the virtual shared expert may encode knowledge broadly shared by all experts.

Figure 9 depicts the resulting layer-0 projection of $W_o$ in our 1B-parameter pretrained UoE. The projections from other layers exhibit similar patterns.

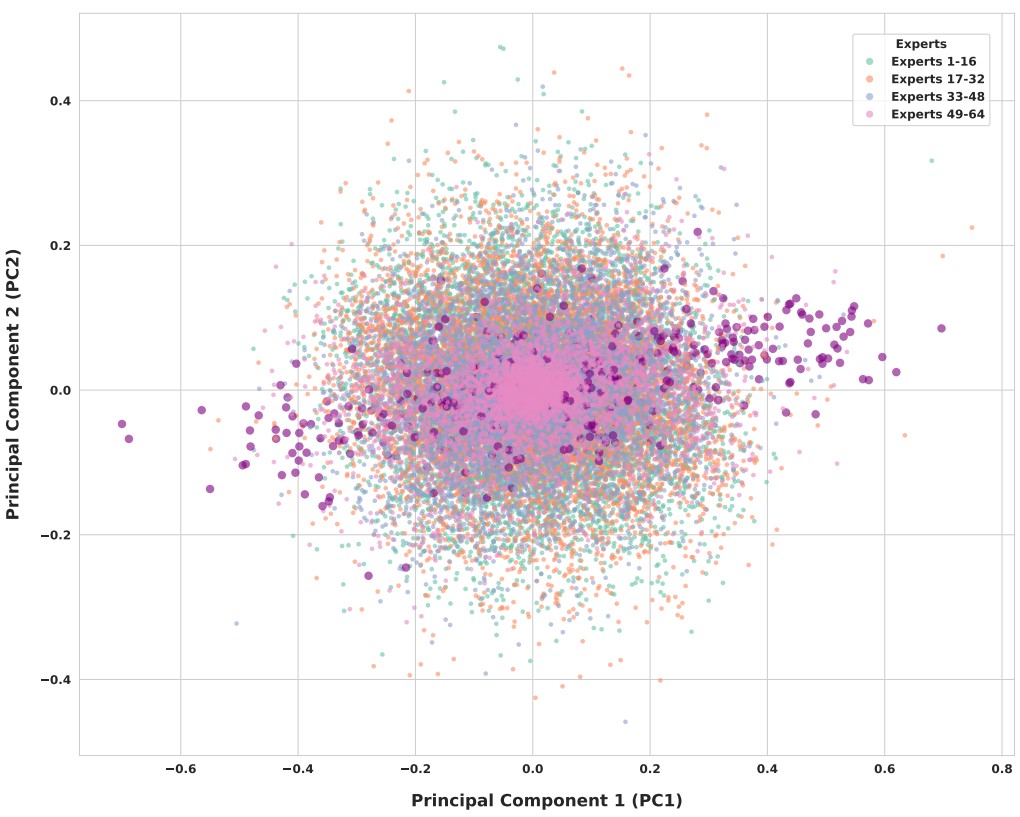

Figure 9: The Principal Component Analyis (PCA) projections of the experts output matrices weights in pre-trained UoE onto the first two principal components ($PC_1$ and $PC_2$), highlighting the routing neurons weights in bold purple. As seen in the plot, these weights project heavily onto the dominant principal components of the full expert $W_o$ matrix. This supports the idea that the virtual shared expert captures knowledge common to all experts.

## G    THE USE OF LARGE LANGUAGE MODELS (LLMS)

This paper employed an LLM solely to refine our manually written draft, including improving word choice, grammar correctness, and sentence fluency.

```python
class MoE(nn.Module):
    def __init__(self, args):
        super().__init__(args)
        self.experts = ParallelMLP(args)
        self.shared_expert = MLP(args)

    def forward(self, x):  # x: [seqlen * bs, hidden_size]
        return self.moe_forward(x) if self.training else self.moe_infer(x)

    def moe_forward(self, x):
        indices = torch.arange(self.N[:, None]) * self.d + \
            torch.arange(self.N_s)[None, :].view(-1)

        wg_ = self.experts.wg[indices]
        wp_ = self.experts.wp[indices]
        wo_ = self.experts.wg[indices]

        expert_acts = F.silu(torch.mm(x, wg_.T)) * torch.mm(x, wp_.T)
        out = torch.mm(expert_acts, wo_)

        expert_acts = expert_acts.view(-1, self.num_experts, self.N_s)
        logits = torch.norm(expert_acts, p=2, dim=-1)
        expert_weights, top_experts = torch.topk(logits, k=self.K, dim=-1)
        expert_weights = expert_weights.softmax(-1, dtype=torch.float32)

        return out + self.experts(x, expert_weights, top_experts)

    @torch.no_grad()
    def moe_infer(self, x):
        # repacking the routing neurons into a virutal shared expert
        if not self.initialized:
            self.create_virutal_shared_expert_weights()
            self.initialized = True

        expert_acts = F.silu(self.shared_expert.wg(x)) * self.shared_expert.wp(x)
        out = self.shared_expert.wo(expert_acts)

        expert_acts = expert_acts.view(-1, self.N, self.N_s)
        logits = torch.norm(expert_acts, p=2, dim=-1)
        expert_weights, top_experts = torch.topk(logits, k=self.K, dim=-1)
        expert_weights = expert_weights.softmax(-1, dtype=torch.float32)

        return out + self.experts(x, expert_weights, top_experts)

    def create_virutal_shared_expert_weights(self):
        self.shared_expert.wg.weight.copy_(
            self.experts.wg.weight.view(
                self.N, self.d , self.D
            )[:, :self.N_s, :].reshape(self.d, self.D)
        )
        self.shared_expert.wp.weight.copy_(
            self.experts.wp.weight.view(
                self.N, self.d , self.D
            )[:, :self.N_s, :].reshape(self.d, self.D)
        )
        self.shared_expert.wo.weight.copy_(
            self.experts.wo.weight.view(
                self.N, self.d , self.D
            )[:, :self.N_s, :].reshape(self.d, self.D)
        )
```

Figure 10: Pseudo code for UoE implementation in PyTorch.

