# OpenReview forum: "Union-of-Experts: Experts in Mixture-of-Experts are Secretly Routers"
_ICLR.cc/2026/Conference — ICLR 2026 Conference Withdrawn Submission_

### Official Review · Reviewer_AGhg · 2025-10-18

**Soundness:** 2
**Presentation:** 3
**Contribution:** 2
**Rating:** 6
**Confidence:** 3

**Summary:**

This paper proposes a new dynamic architecture for LLMs, termed Union-of-Experts (UoE), designed to address the disconnect between the routing module and expert capabilities in traditional MoEs while overcoming the computational and memory efficiency bottlenecks of Autonomy-of-Experts (AoE). UoE use some neurons within each expert are designated as routing neurons, whose activation strength is used to determine the expert's suitability for processing the current input. The routing neurons of all experts collectively form a virtual shared expert, whose output is not only used for routing decisions but also directly contributes to the final prediction, thus avoiding computational waste. UoE consistently outperforms both MoE and AoE on multiple downstream tasks using pre-trained language models with 1B and 3B parameter sizes. It also achieves a training throughput comparable to MoE.

**Strengths:**

1. The authors combines routing neurons into "virtual shared experts" to reuse routing signals and avoid redundant computation in AoE.

2. Detailed ablation experiments are provided to verify the effectiveness of the virtual shared expert and routing neuron design.

**Weaknesses:**

1. The experiment lacks implementation details.

2. The scale of the experiment is relatively limited, and both the data and model scale are relatively limited.

**Questions:**

1. The inference efficiency of MoE is significantly affected by its implementation method. The author should explain how MoE is implemented.

2. In light of the constrained experimental scale, the authors ought to illustrate how model capability evolves with increasing training-data size, thereby revealing the method’s potential at truly large scales.

3. How efficient is the inference of the proposed method?

---

> ### Author Response · Authors · 2025-11-24
> **Response to Reviewer AGhg**
>
> We sincerely thank you for your constructive suggestions and valuable comments! We hope our rebuttal helps address your concerns.
>
> ---
> **W1 & Q1: More detailed implementation details.**
>
> In the initial submission, we provided the code implementation as well as the necessary implementation details in Section 4.1.
> We appreciate the reviewer’s suggestion for improving the clarity of our presentation.
> In the revised version, we have included additional implementation details (please refer to the Appendix).
>
> ---
> **W2 & Q2: Concerns about the scalibility of UoE.**
>
> Thank you for your suggestion. I understand the reviewer’s concerns regarding the scalability of UoE, and we would like to provide some clarification.
>
> As an exploratory study on MoE routing, we align our model scale and training tokens with prior work [1] (AoE, ICML 2025) as closely as possible.
> Under our current computational budget, pre-training a 3B UoE model from scratch requires roughly 2000 GPU hours, while conducting a full comparison would exceed 7500 GPU hours. Given the limitations in time and compute, we have made our best effort to scale UoE models within our constraints.
>
> We are currently pre-training a larger 7B UoE model.
> We will update the results for larger-scale UoE training and share them with you as soon as possible.
>
> [1] Autonomy-of-Experts Models, ICML 2025
>
> ---
> **Q3: Empirical evidence for the inference efficiency of UoE.**
>
> Thank you for the insightful question. We have added an inference efficiency comparison between UoE and MoE in our revised version.
>
> Specifically, we implement a generation pipeline using HuggingFace and conduct a detailed breakdown analysis of UoE’s inference efficiency, benchmarking both peak memory usage and end-to-end generation throughput. All experiments use 256 random input tokens and span a range of generation lengths and batch-size settings.
>
> We summarize the experimental results below:
> | Configuration | | TP. (token/s)  | Mem. (GB) | |
> | :---: | :---: | :---: | :---: | :---: |
> | **Model** | **BS** | **256** | **1024** | **4096** |
> | MoE | 1 | 35.71 (2.15) | 35.84 (2.15) | 35.81 (2.15) |
> | UoE | 1 | 35.84 (2.15) | 35.98 (2.15) | 35.89 (2.15) |
> | MoE | 4 | 141.99 (2.21) | 141.38 (2.21) | 141.37 (2.21) |
> | UoE | 4 | 140.83 (2.21) | 141.00 (2.21) | 141.46 (2.21) |
> | MoE | 16 | 561.82 (2.48) | 560.93 (2.48) | 559.97 (2.48) |
> | UoE | 16 | 549.91 (2.47) | 551.39 (2.47) | 551.93 (2.47) |
>
> These results show that the computational overhead of UoE is nearly identical to that of MoE.
> Additional details can be found in our revised paper.

---

### Official Review · Reviewer_HmyS · 2025-10-28

**Soundness:** 2
**Presentation:** 3
**Contribution:** 3
**Rating:** 4
**Confidence:** 4

**Summary:**

This paper introduces Union-of-Experts (UoE), a novel Mixture-of-Experts (MoE) architecture. UoE addresses two key limitations: the router in standard MoE is unaware of expert capabilities, and the Autonomy-of-Experts (AoE) incurs significant computational overhead. UoE replaces the standard router by utilizing a small subset of "routing neurons" within each expert's weight matrix to parameterize the routing function. This design enables capabilities-aware routing without the complexity or cost of AoE's low-rank factorization. Furthermore, these routing neurons are collectively unified to form a virtual shared expert, ensuring their activations contribute to the final output and preventing computational waste. Experiments show that UoE consistently outperforms MoE and AoE baselines while maintaining comparable efficiency.

**Strengths:**

The motivation is clear: achieve expert autonomy by reducing the overhead of AoE while eliminating the router in MoE. The idea is easy to grasp, requires minimal architectural changes, and incurs little overhead. By using a small subset of routing neurons and integrating their computation as a virtual shared expert, UoE ensures efficient resource utilization and incurs overhead nearly identical to standard MoE while delivering superior performance.

**Weaknesses:**

The primary weaknesses revolve around a lack of detailed implementation specifics and insufficient empirical evidence to support key efficiency claims, as detailed in the questions below.

Minor comments:

* Fig. 2 is taken directly from the AoE paper, the authors may need to redraw the figure for the camera-ready version to prevent potential copyright issues.

**Questions:**

1. Is the efficiency dilemma presented in Figure 3, which plots FLOPs and Peak Memory as a function of factorization rank $r$, a purely theoretical estimate based on the derived equations (Eq. 4 and Eq. 5), or is it based on empirical profiling numbers from real experiments? If it's a theoretical estimate, this should be clearly emphasized in Section 3.1 or in the figure caption.

2. In Table 2, the achieved training TFLOPS for UoE ($86.51$) is notably lower than that of MoE ($90.40$). Could the authors elaborate on the implementation details in the Appendix to explain this difference? For example, during training, are the router neurons computed first, and then the final expert values re-computed? Will the router neurons' input-to-output path be calculated twice? Or, is it computed only once and then the output is efficiently merged with the remaining neurons' output at the code level? This isn't held against the authors, but some explanations would be highly preferred. It seems the router neurons' computation path might need to be utilized twice during training, potentially explaining the lower TFLOPS compared to MoE.

3. Why do UoE models have a higher throughput but lower TFLOPS compared to MoE in Table 2?

4. The claim that "UoE incurs computational overhead that is nearly identical to MoE at inference time"  lacks experimental validation. Could the authors provide an inference profiling analysis (e.g., inference TFLOPS, peak memory, and throughput) similar to Table 2 to quantitatively support this claim? In addition, a dedicated section in the Appendix detailing the implementation for efficient training/inference of UoE (e.g., regarding the computation of router neurons) would be highly beneficial.

5. The title "Union-of-Experts: Experts in Mixture-of-Experts are Secretly Routers" is a bit misleading. The core mechanism involves explicitly constraining a small subset of neurons to represent routing information during training. The current title seems to suggest that a standard expert's full output could be used as a router directly, without training-time modifications. If possible, could the authors revise the title to more accurately reflect the main contribution, which is the use of specialized routing neurons and the virtual shared expert?

---

> ### Author Response · Authors · 2025-11-24
> **Response to Reviewer HmyS**
>
> We sincerely appreciate your valuable comments! Hope our response will help address your concerns.
>
> ---
> **Q1: Is the efficiency dilemma presented in Figure 3 a theoretical estimate?**
>
> Yes, and we have revised both the figure caption and the content in Section 3.1 accordingly.
> We appreciate your helpful suggestions for improving the presentation of our paper.
>
> ---
> **Q2: Additonal implementation details of UoE.**
>
> Thank you for the suggestion.
> In the supplementary material of our initial submission, we provided the code implementation of UoE.
> In this revision, we have updated the appendix to include a more detailed discussion of the implementation, along with pseudocode in pytorch to better illustrate the core algorithm.
>
> Below, we respond to your concerns regarding the implementation of UoE:
>
> > Does UoE recompute the routing neurons during training?
>
> Yes, and we want to clarify the reason.
> Our MoE implementation is built on MegaBlocks [1], which requires the FFN hidden size to be a multiple of 128 to accelerate sparse matrix computation.
> Isolating the routing neurons and merging them back would break this hidden-size constraint and is not efficient in practice.
> This explains why these routing neurons are forwarded twice in our design.
> We will include this part of discussion in our future revision.
>
> We again thank the reviewer for the helpful feedback that improved the presentation of our paper.
>
> [1] https://github.com/databricks/megablocks
>
> ---
>
> **Q3: Why UoE have a higher throughput compared to MoE ?**
>
> Your observation is very insightful.
> We would like to clarify that Table 2 reports the tflops at training step 1, as the tflops of a freshly initialized XoE model better reflects the algorithmic complexity.
> For throughput, we report the overall tps measured during training.
>
> It is worth noting that tflops is only one factor influencing the efficiency of XoE model training.
> Because UoE achieves more balanced expert utilization than standard MoE, it attains better end-to-end throughput in practice, as is shown in Table 2.
>
> We sincerely thank the reviewer for this insightful question, which helped refine the presentation of our paper. We will incorporate this discussion in a future revision.
>
> ---
> **Q4: Empirical evidence for the inference efficiency of UoE.**
>
> Thank you for the insightful question and we have included an inference efficency comparison for UoE and MoE in our revised version.
>
> Specifically, we implement a generation pipeline built on HuggingFace.
> We conduct a detailed breakdown analysis of UoE’s inference efficiency, benchmarking both peak memory usage and end-to-end generation throughput.
> All experiments use 256 random input tokens and cover a range of generation lengths and batch-size configurations.
>
> We summarize the experimetal results below:
> | Configuration | | TP. (token/s)  | Mem. (GB) | |
> | :---: | :---: | :---: | :---: | :---: |
> | **Model** | **BS** | **256** | **1024** | **4096** |
> | MoE | 1 | 35.71 (2.15) | 35.84 (2.15) | 35.81 (2.15) |
> | UoE | 1 | 35.84 (2.15) | 35.98 (2.15) | 35.89 (2.15) |
> | MoE | 4 | 141.99 (2.21) | 141.38 (2.21) | 141.37 (2.21) |
> | UoE | 4 | 140.83 (2.21) | 141.00 (2.21) | 141.46 (2.21) |
> | MoE | 16 | 561.82 (2.48) | 560.93 (2.48) | 559.97 (2.48) |
> | UoE | 16 | 549.91 (2.47) | 551.39 (2.47) | 551.93 (2.47) |
>
> We argue that the computation overhead of UoE is nearly identical to that of MoE.
> More details can be found in our revised paper.
>
>
> ---
> **Q5: A clarification of the title name**
>
> Thank you for the suggestion and we want of clarify that.
> Our intention was to convey that the MoE router is implicitly encoded in the expert weights, as part of the UoE expert weights is used for routing.
> We understand your concerns and appreciate the feedback, and we will refine the title in future revisions.

---

### Official Review · Reviewer_HdQY · 2025-10-29

**Soundness:** 2
**Presentation:** 2
**Contribution:** 2
**Rating:** 4
**Confidence:** 4

**Summary:**

This paper proposes Union-of-Experts (UoE), a new Mixture-of-Experts (MoE) variant that unifies routing and expert computation to improve both efficiency and expert autonomy. The key insight is that a small subset of neurons—termed routing neurons—within each expert can represent the expert’s activation behavior and be used for routing decisions based on their activation norms. By consolidating these routing neurons from all experts into a virtual shared expert, UoE eliminates redundant computations present in Autonomy-of-Experts (AoE) while preserving expert autonomy.

The paper provides both theoretical analysis (showing how AoE’s low-rank factorization leads to inefficiency) and empirical validation, pretraining models up to 3B parameters on large-scale corpora. Experiments on multiple benchmarks (ARC, PIQA, HellaSwag, etc.) show that UoE outperforms both MoE and AoE in accuracy while maintaining computational efficiency comparable to MoE. UoE also achieves better expert load balancing and scalability as model size increases. Ablation studies confirm the importance of the virtual shared expert and the routing neuron mechanism.

**Strengths:**

1. Well-motivated and clearly articulated problem: bridging the gap between routing decisions and expert capabilities.
2. Elegant design that combines autonomy-based routing with MoE efficiency.
3. Extensive experimental validation (1B–3B models) with consistent improvements.
4. Solid analysis on computation–memory trade-offs and load balancing.

**Weaknesses:**

1.The core claim—that external routers are unaware of expert capabilities—is not empirically validated, e.g., consistency analysis between router choices and optimal expert matches.

2.The key assumption that a fixed subset of “routing neurons” can represent full expert behavior is supported mainly by empirical correlation, without rigorous theoretical or mechanistic explanation.

3.The study only validates UoE on models up to 3B parameters, lacking comparison with larger-scale MoE architectures (e.g., DeepSeek-MoE) and raising concerns about scalability.

4.Experiments are mostly conducted on commonsense reasoning and NLU tasks. Broader capabilities such as code generation and mathematical reasoning are remain unexamined.

5.While improved balance is shown, the underlying reasons are not explored, nor is its potential trade-off with routing quality.

6.It is unclear whether these neurons learn general-purpose functions or merely serve as routing proxies, weakening the design's motivation.

7.The “virtual shared expert” is described as abstract during training and materialized at inference, complicating reproducibility and clear implementation.

**Questions:**

These questions correspond to the weaknesses listed above:

1.Have you empirically verified that external routers are misaligned with expert capabilities, e.g., through consistency analysis?

2.What justifies that a fixed subset of neurons can represent full expert behavior beyond empirical correlation?

3.How does UoE scale to larger MoE models (e.g., 10B–72B) and industrial training settings?

4.Can the proposed routing mechanism generalize to other capabilities such as code or multimodal reasoning?

5.Why does UoE achieve better load balance, and is there a trade-off with routing quality?

6.Do routing neurons learn general-purpose representations or act purely as routing proxies?

7.The description of the “virtual shared expert” is unclear — could the authors clarify its implementation and how it differs between training and inference?

---

> ### Author Response · Authors · 2025-11-24
> **Response to Reviewer HdQY (0)**
>
> We sincerely appreciate your valuable comments and hope our responses address your concerns.
>
> ---
> **W1 & Q1: Empirical evidence suggests external routers are unaware of expert capabilities**
>
> We run experiments on 16B DeepSeekMoE [1] to illustrate the separation between router decisions and their awareness of expert capabilities.
>
> Specifically, we modify the routing logic by selecting the top-k experts according to the L2-norm of expert activations (SiLU(wg) * wp) instead of the router logits.
> The router logits are still used to compute the expert weights.
>
> The performance comparison on MMLU and HellaSwag is summarized below:
> | Config | MMLU | HellaSwag |
> | - | - |- |
> | DeepSeekMoE | 44.8 | 77.3 |
> | Expert Act | 39.8 | 74.1 |
>
> Consistent observations are reported by Lv et al [2], further supporting this separation.
>
> We hope these results help clarify your concerns.
>
>
> [1] https://huggingface.co/deepseek-ai/deepseek-moe-16b-base
>
> [2] Autonomy-of-Experts Models, ICML 2025
>
> ---
> **W2 & Q2: An mechanism explanation of how routing neurons represent full expert behavior**
>
> Thank you for the insightful question.
> In the updated version of our work, we present a theoretical explanation detailing how the routing neurons are able to represent the full expert behavior.
>
> Specifically, we observe that these routing neurons tend to align with the principal singular vectors of the expert weight matrix.
> This alignment allows them to generate an accurate approximation of the full expert's activations.
> This behavior is less pronounced in the router weights of MoE models.
> We attribute the emergence of this specific capability to the expert-autonomy training dynamics utilized in UoE and AoE models.
>
> More detail can be found in Appendix D.
>
>
> ---
> **W3 & Q3: Concerns about the scalibility of UoE**
>
> We appreciate the reviewer’s recognition of our efforts to scale the model.
> As an exploratory study on MoE routing, we align our model scale and training tokens with prior work [1] (AoE, ICML 2025) as closely as possible.
>
> Under our computational resources, pre-training a 3B UoE model from scratch requires roughly **2000 GPU hours**, and a full comparison would **exceed 7500 GPU hours**. Given the limitations in time and computational resources, we have made our best efforts to scale UoE models.
>
> We are currently pre-training a larger UoE model (7B) and we leave the pre-training of 10B+ MoE models to future work due to resource constraints.
> We will update the results for larger-scale UoE training and discuss them with you **as soon as possible**.
>
> [1] Autonomy-of-Experts Models, ICML 2025
>
> ---
> **W4 & Q4: Additional evaluations on multimodal reasoning and code**
>
> > UoE's performance on multi-modal reasoning.
>
> We appreciate the reviewer’s suggestion. However, this work primarily focuses on language-model pre-training.
> To our knowledge, multimodal pre-training (e.g., VLLM) differs substantially from pure LLM pre-training, both in model architecture and training methodology.
>
> Given these differences, we are afraid we will not include this direction in our current discussion.
>
> > Code generation & Mathematical Reasoing.
>
> We want to clarify that the pre-trained base language models used in our experiments (trained on 100B tokens with at most 3B parameters) have not yet developed meaningful code-generation or mathematical-reasoning abilities. Therefore, evaluating them on such benchmarks would not be informative at this stage.
> Similar setups appear in DeepSeek-MoE [1], where results for small models on code and math are also not reported.
>
> We again acknowledge the value of exploring these capabilities, and they may guide future extensions of our work.
>
> [1] DeepSeekMoE: Towards Ultimate Expert Specialization in Mixture-of-Experts Language Models
>
> ---
> **W5 & Q5: Explanation for why UoE achieves better load balance ?**
>
> We are pleased that the reviewer acknowledges UoE’s improved load-balancing behavior, and we respond to the concerns as follows.
>
> > Why does UoE achieve better load balance ?
>
> Lv et al.[1] show that AoE delivers superior load balance compared to traditional MoE, together with higher routing quality (AoE achieves lower pre-training loss and better downstream performance).
>
> UoE adopts the expert-autonomy routing strategy in AoE, and we observe similar behavior (better loading balance & routing quality) in UoE.
> We hypothesize that these benefits stem from routing based on the L2 norm of activations, which differs fundamentally from the router in traditonal MoE.
>
> We are still activately working toward a mechanistic explanation for that, thank you for your question!
>
> > Is there a trade-off between load balance and routing quality.
>
> Our evaluation of UoE on downstream tasks (showing both better performance and better load balance) indicates that there is no inherent trade-off.
>
> [1] Autonomy-of-Experts Models, ICML 2025

---

> ### Author Response · Authors · 2025-11-24
> **Response to Reviewer HdQY (1)**
>
> ---
> **W6 & Q6: Do routing neurons learn general representations ?**
>
> Thank you for the insightful question; it inspires us to take a deeper insight into the routing neurons, and here are our findings:
>
> We apply PCA to the output weight matrices ($W_o$) of the UoE experts and visualize their projections onto the first two principal components.
> We find that the routing neurons tend to concentrate along the dominant component and exhibit pronounced projection magnitudes.
> This pattern strongly suggests that the routing neurons may be encoding knowledge or features shared among the different experts within the MoE layers.
>
> We have included the detailed discussion of this finding in Appendix D.
>
> ---
> **W7 & Q7: Additional clarification on the virtual shared expert.**
>
> We have updated the paper to clarify the concept of the virtual shared expert.
>
> The minor difference between the virtual shared expert during training and inference is that the routing neurons — which are initially distributed across different experts — are repacked into a layout that conforms to the standard shared expert structure.
>
> For implementation clarity, we have also provided Python pseudocode illustrating this process in the Appendix.
> Further technical details are available in our code repository.

---

### Official Review · Reviewer_yzUJ · 2025-11-01

**Soundness:** 3
**Presentation:** 3
**Contribution:** 3
**Rating:** 4
**Confidence:** 3

**Summary:**

This paper introduces a novel Mixture-of-Experts architecture that addresses the fundamental limitation where traditional routers operate externally to experts and remain unaware of their capabilities. The key innovation is demonstrating that a small subset of "routing neurons" ($N_s \ll D$) within each expert can simultaneously perform two functions: their activation norms enable autonomous expert selection based on capability-input matching, and they collectively form a "virtual shared expert" that contributes to the final output, eliminating the computational waste of prior autonomous approaches. Experiments on language models up to 3B parameters show UoE consistently outperforms MoE and AoE baselines across 8 benchmarks while achieving 19.8% higher training throughput than AoE and maintaining computational costs identical to standard MoE, with additional benefits including improved load balancing without auxiliary losses.

**Strengths:**

- The paper presents an interesting architectural contribution where a small subset of neurons within each expert serves the dual function of routing decisions and output contribution, offering a novel perspective on integrating expert selection with expert computation that differs from conventional external router approaches.

- The experimental evaluation provides thorough efficiency comparisons demonstrating that UoE achieves notable throughput improvements over AoE (19.8%) while maintaining computational costs comparable to standard MoE (Table 2), and the load balancing analysis (Figure 6, Table 3) reveals encouraging improvements across tested configurations.

- The paper effectively communicates the core limitation of routers being decoupled from expert capabilities, employs helpful visual illustrations (Figures 1 and 4) that clarify the architectural distinctions, and maintains consistent mathematical formalism (Equations 1-6) throughout the exposition to support reader understanding.

**Weaknesses:**

- The paper lacks rigorous theoretical justification for why the first Ns neurons can spontaneously learn to represent the entire expert's activation patterns. While the authors claim this correlation "spontaneously emerges during training" (Section 3.2), they provide no mathematical framework or analysis explaining why this should occur. The statement that "the selection of routing neurons proves highly flexible" is supported only by empirical observations relegated to Appendix B, without exploring what properties make certain neuron subsets more effective than others. A more principled approach would derive conditions under which routing neurons can provably approximate full expert activations, perhaps through analysis of gradient flow or information-theoretic bounds. Without this foundation, the core mechanism appears somewhat arbitrary and the generalizability to different architectures remains uncertain.

- While the paper presents results up to 3B parameters, modern production MoE models operate at scales of hundreds of billions to trillions of parameters (e.g., GPT-4, Mixtral-8x7B). The largest model tested (3B) is too small to convincingly demonstrate that UoE's design principles will hold at scale. Specifically, it remains unclear whether: (a) the routing neuron correlation property persists as expert capacity grows, (b) the load balancing advantages maintain when N (number of experts) increases to 64 or 256 as in recent models, and (c) the memory savings remain significant when model width increases substantially. The authors acknowledge that "wide models with large d and D" face training instability (Section 3.1) but do not thoroughly investigate this limitation. Experiments on at least a 7B or 13B model would significantly strengthen the scalability claims.

- The conceptual contribution of the "virtual shared expert" lacks depth in several aspects. First, the paper does not adequately explain why routing neurons should serve the dual purpose of routing and shared expert computation—this seems more like an engineering convenience than a principled design choice. Second, the ablation study (Table 4, configurations 1-3) shows that removing the shared expert causes performance degradation, but this is expected for any MoE with shared experts and does not validate that routing neurons specifically are the right choice for this role. Third, the paper claims the virtual shared expert "consolidates common capabilities" but provides no analysis of what capabilities it actually learns compared to routed experts or traditional shared experts. A more thorough investigation would include: visualization of what features the virtual shared expert captures, analysis of its activation patterns across different token types, and comparison of learned representations with those of conventional shared experts.

- The evaluation is limited to 8 standard benchmarks that primarily test knowledge recall and basic reasoning (ARC, PIQA, HellaSwag, etc.), but lacks evaluation on: (a) generation quality metrics (perplexity on diverse corpora, human evaluation of generated text), (b) instruction-following capabilities, (c) domain-specific tasks where expert specialization matters most (e.g., code generation, mathematical reasoning, multilingual understanding), and (d) long-context understanding where expert selection patterns might differ significantly. Furthermore, all benchmarks are English-only, raising questions about whether UoE's routing mechanism generalizes across languages. The lack of diverse evaluation makes it difficult to assess whether UoE's improvements are robust across different use cases or merely artifacts of the specific benchmark selection.

- Several key design decisions lack proper justification or ablation studies: (a) Why is Ns = round(D/K) the optimal choice? The paper only ablates doubling Ns but doesn't explore the full spectrum or provide sensitivity analysis. (b) Why are routing neurons placed at the "first Ns neurons" rather than being learned positions or distributed throughout the weight matrix? (c) The choice of L2-norm for measuring activation intensity is tested against only 3 alternatives (Table 4, configs 5-7), but other options like L1-norm, max activation, or learned weighted combinations are not explored. (d) The paper doesn't ablate the impact of the SiLU activation function specifically in the routing computation. (e) No analysis of how performance varies with different values of K (number of activated experts) beyond the tested configurations. A more comprehensive ablation study would systematically explore the design space and provide guidance for adapting UoE to different scenarios.

**Questions:**

- Could you provide a more rigorous theoretical or mathematical explanation for why routing neurons (specifically the first $N_s$ neurons) can effectively capture the match between expert capabilities and input tokens? While Section 3.2 states this correlation "spontaneously emerges during training," the mechanism remains unclear. Specifically: (a) What gradient dynamics or optimization properties lead to this emergent behavior? (b) Are there any information-theoretic bounds or approximation guarantees that justify using $N_s \ll D$ neurons? (c) Under what conditions might this correlation fail to emerge? A theoretical framework would strengthen the core contribution and help practitioners understand when UoE might underperform.

- Given that modern production MoE models operate at scales of 7B-100B+ parameters with 8-64+ experts, how confident are you that UoE's properties will hold at these scales? Could you discuss: (a) Have you conducted any preliminary experiments or analyses suggesting the routing neuron correlation persists as expert capacity grows significantly? (b) How does load balancing and routing quality change when $N$ (number of experts) increases to 16, 32, or 64? (c) What are the specific training instabilities you observed with "wide models" mentioned in Section 3.1, and what architectural modifications might address them? Understanding these scalability characteristics is crucial for assessing UoE's practical applicability.

---

> ### Author Response · Authors · 2025-11-24
> **Response to Reviewer yzUJ (0)**
>
> Thank you for valuable feedbacks! Hope our explanation helps address your concerns.
>
> ---
> **W1 & Q1: Provide a theoretical explanation of how routing neurons model the full activation patterns of experts.**
>
> Thank you for the insightful question; it offered us a fresh perspective on understanding UoE and improving the presentation of the paper.
>
> > What properties make certain neurons function as the routing neurons ?
>
> In the updated version, we introduce a theoretical explanation of how the routing neurons capture the activation patterns of the full expert.
> We observe that these neurons tend to align with the principal singular vectors of the expert weight matrix, thus enabling accurate approximation of expert activations
>
> Such behavior is not s as pronounced in the router weights of standard MoE models
> and we attribute the emergence of this capability to the expert-autonomy training dynamics in UoE and AoE.
>
> > Justification for the chosen number of routing neurons.
>
> This analysis also offers an information-theoretic justification for the chosen number of routing neurons:
> MoE expert weights are effectively low-rank and can be approximated well using only the leading singular vectors [1-2].
>
> Please refer to Appendix D for further details.
>
> [1] Autonomy-of-Experts Models, ICML 2025
>
> [2] Delta Decompression for MoE-based LLMs Compression, ICML 2025
>
> ---
> **W2 & Q2: Concerns over UoE’s effectiveness at larger scales.**
>
> > Experiments on 7B or larger models to substantiate the scabitlity claims.
>
> We appreciate the reviewer’s recognition of our efforts to scale the model.
> As an exploratory study on MoE routing, we align our model scale and training tokens with prior work [1] (AoE, ICML 2025) as closely as possible.
> We believe the training costs are on par with those of recent architecture exploration studies [1–2].
> Given the limitations in time and computational resources, we have made our best efforts to scale UoE models.
>
> **We are currently pre-training a larger UoE model (7B).**
> Results from the 7B UoE training, along with discussions of its properties (routing neuron effectiveness, load balance, etc.), will be shared **as soon as possible**.
>
> [1] Autonomy-of-Experts Models, ICML 2025
>
> [2] MoH: Multi-Head Attention as Mixture-of-Head Attention, ICML 2025
>
> > Can load balance advantages remain when N increase ?
>
> Across experiments, we set the number of routed experts as 64 to align with current mainstream configurations [3–4].
> We believe that N=64 already satisfies your requirements in Q2 and
> demonstrates UoE's load-balancing advantages with a large number of routed experts.
>
> [1] DeepSeekMoE: Towards Ultimate Expert Specialization in
> Mixture-of-Experts Language Models
>
> [2] OLMoE: Open Mixture-of-Experts Language Models
>
> > Clarification of the training instabilities in Section 3.1
>
> We would like to clarify that the observed training instability occurs when reproducing AoE rather than UoE;
> in this setting, a relatively small r can lead to training collapse even in a 1B setup.
>
> ---
> **W3: Concerns about the design of the virtual shared expert.**
>
> Thank you for the suggestions. We have added additional experimental results, and hope this can help address your concerns.
>
> > (a) Analysis of the parametric knowledge encoded in the virtual shared expert.
>
> We apply PCA to the $W_o$ matrices of UoE’s experts and visualize their projections onto the first two principal components.
> The routing neurons are found to concentrate along the dominant component with pronounced projection magnitudes.
> This pattern suggests that the routing neurons may encode knowledge shared across experts in the MoE layers.
>
> We have included a detailed discussion of this finding in Appendix D.
>
> > (b) Additional ablation study of the virtual shared expert.
>
> We perform an additional ablation study to evaluate the impact of the virtual shared expert.
> In this configuration, the routing neurons handle expert routing, with an additional shared expert included on top of Configuration 3.
>
> The average performance across the eight datasets is nearly identical to that of UoE, suggesting that the virtual shared expert can serve a role similar to that of a conventional shared expert.
>
> | Configuration | ARC-E | PIQA | HELLA | SCIQ | WINO | MNLI | QNLI | RTE | AVG. |
> | - | -| - | -| - | - | -| - | - | - |
> | UoE | 63.09 | 69.64 | 37.07 | 82.40 | 52.88 | 33.89 | 50.05 | 51.50 | 55.07 |
> | Ablation | 64.56 | 69.10 | 36.78 | 81.70 | 53.67 | 32.93 | 50.78 | 50.82 | 55.04 |
>
> > The virtual shared expert is an engineering convenience, not a principled design decision.
>
> The virtual shared expert is intended initially to reuse intermediate computations, thus mitigating the routing overhead introduced by expert autonomy.
> Together with the updated experimental results, the virtual shared expert is an effective solution, functioning as a consolidator of shared knowledge while facilitating activation value reuse.

---

> ### Author Response · Authors · 2025-11-24
> **Response to Reviewer yzUJ**
>
> - **W4: Additional evaluations on a broader set of benchmarks**
>
> Thank you for your suggestion.
> However, we will not include certain benchmarks, and we would like to clarify this.
>
> > Instruction-following
>
> We appreciate the reviewer’s suggestion. However, this work primarily focuses on language model pre-training. UoE and the baseline methods are base models **without** SFT or alignment with human preferences, so evaluating their instruction-following ability is not meaningful.
> Our setup follows [1], [2] and to my knowledge, is also a default setup in exploratory work on pre-training new architectures.
>
> > Human evaluation of generated text, Long-context understanding, Code generation, Mathematical reasoning & Multilingual understanding ?
>
> We want to clarify that the pre-trained base language models used in our experiments (trained on 100B tokens with at most 3B parameters) have not acquired the above capabilities during pre-training, so evaluating them on these tasks would not be meaningful.
> Similar observations can be found in [2], where results for small models on code and math are also not reported.
>
> We again thank the reviewer for bringing these capabilities to our attention, as they may guide future extensions of our work.
>
> [1] Autonomy-of-Experts Models, ICML 2025
>
> [2] DeepSeekMoE: Towards Ultimate Expert Specialization in
> Mixture-of-Experts Language Models
>
> ---
> **W5: Clarification of some experimental designs in UoE**
> > (a) The choice of N_s in ablation studies
>
> We set N_s to round(D / K) or 2 * round(D / K) for two reasons.
>
> First, we choose N_s as a multiple of round(D / K).
> If N_s were not divisible by round(D / K), we would need to adjust the FFN hidden size to maintain a consistent number of activated parameters.
> An irregular hidden size could lead to inefficient MoE training and a less elegant design, so we keep N_s as a multiple of round(D / K).
>
> Second, since the number of shared experts is usually no more than two, we adopt this standard practice.
>
> > (b) Why are routing neurons placed at the "first Ns neurons"
>
> As stated in Appendix B, we choose to fix a subset of neurons to act as routing functions.
> These routing neurons are trained from scratch alongside the experts to maintain expert autonomy.
> Given this setup, selecting different neurons from the expert weights would only change the weight initialization.
> To simplify implementation, we simply use the first contiguous N_s neurons for expert routing.
>
> > (c) The choice of L2-norm for measuring activation intensity
>
> Lv et al. [1]  conducted preliminary experiments showing that L2-norm is an optimal choice.
> As a consequence, we follow this setup in our work.
>
> [1] Autonomy-of-Experts Models, ICML 2025
>
> > (d) The ablation of SiLU activation function
>
> Thank you for the suggestion.
> We want to clarify that, in Configuration 6 and 7, we have conducted the ablation study of the impact of SiLU function.
>
> > (e) How performance varies with different values of K
>
> Thank you for the suggestion.
> We want to clarify that, we have presented the UoE's performance with diffent K values in Table 1 (K=8 and K=4).

---

> > ### Comment · Reviewer_yzUJ · 2025-11-27
> > **Thanks for your response**
> >
> > Thank you for your detailed response and the additional analyses, particularly regarding the SVD alignment and the PCA visualization of the virtual shared expert. These additions have helped clarify the intuition behind the routing neurons.
> >
> > I have a few follow-up comments and queries:
> >
> > 1.  I appreciate your effort in training a 7B model. Given that the effectiveness of novel MoE routing mechanisms often varies significantly with scale, the results of the 7B experiment are pivotal for substantiating the scalability claims. Looking forward to seeing these results during the discussion period.
> >
> > 2.  The empirical observation that routing neurons align with principal singular vectors is interesting. However, does this alignment remain consistent across all layers (e.g., deeper layers vs. shallow layers) and throughout the entire training process?
> >
> > 3.  I accept your clarification regarding the limitations of evaluating specific capabilities (math/code) on base models trained with 100B tokens.
> >
> > I look forward to your updates.

---

> > > ### Author Response · Authors · 2025-12-03
> > > **Response to Reviewer yzUJ**
> > >
> > > We appreciate your quick response and thank for your understanding in not requiring tests on certain benchmarks.
> > > Below, we address the remaining questions:
> > >
> > > > Does the alignment align with principal singular vectors across layers during training?
> > >
> > > We appreciate the reviewers' interest in this alignment.
> > > We would like to clarify that Figures 7 and 8 in Appendix D already present similarity measurements across different layers, with the y-axis representing layer indices and the x-axis indicating expert indices.
> > > Across all layers, we observe that UoE's routing neurons exhibit a stronger tendency to align with the principal direction compared to the MoE router.
> > > Notably, this alignment is more pronounced in the deep and shallow layers than in the intermediate ones.
> > >
> > > In response, we will refine the text in Appendix D to provide a clearer explanation of this phenomenon and to prevent potential misunderstandings.
> > > We have also added a heatmap from a UoE training checkpoint, which shows a similar pattern.
> > > We will include discussion related in our future revision.
> > >
> > > During the rebuttal phase, we made efforts to obtain the necessary resources to complete the training of UoE-7B.
> > > However, due to resource limitations, we are currently unable to provide the detailed additional experimental results.
> > > We hope to include them in our future revision.
> > >
> > > We sincerely thank the reviewers for their constructive discussions during the review process, which have been invaluable in helping us improve the presentation.
> > > Once again, we are deeply grateful for your time and efforts.

---

### Note · Authors · 2025-12-03

**Comment:**

I have read and agree with the venue's withdrawal policy on behalf of myself and my co-authors.

**Withdrawal Confirmation:**

I have read and agree with the venue's withdrawal policy on behalf of myself and my co-authors.